



# Terrestrial Cosmogenic Nuclide depth profiles used to infer changes in Holocene glacier cover, Vintage Peak, Southern Coast Mountains, British Columbia

Adam C. Hawkins[1], Brent M. Goehring[2], Brian Menounos[1,3]

[1]Department of Geography, Earth, and Environmental Sciences, University of Northern British Columbia, Prince George, BC, Canada, V2N 4Z9
[2]Los Alamos National Laboratory, Los Alamos, 87545, USA
[3]Hakai Institute, Campbell River, V9W 2C7, Canada

*Correspondence to:* Adam C. Hawkins (adam.hawkins@unbc.ca)

**Abstract.**

The majority of glaciers in North America reached their maximum Holocene downvalley positions during the Little Ice Age (1300-1850 CE), and in most cases, this expansion also destroyed earlier evidence of glacier activity. Substantial retreat in the 20th and early 21st centuries exposed bedrock that fronts many glaciers that may record early-to-mid Holocene exposure and later burial by ice which can be elucidated using multiple-nuclide cosmogenic surface exposure dating. Furthermore, cores of bedrock allow the measurement of cosmogenic nuclide depth profiles to better constrain potential exposure and burial histories. We collected four bedrock surface samples for $^{10}$Be and $^{14}$C surface exposure dating and shallow bedrock cores from Vintage Peak, in the southern Coast Mountains of British Columbia, Canada. We apply a Monte Carlo approach to generate combinations of exposure and burial duration that can explain our data. Vintage Peak became uncovered by the Cordilleran Ice Sheet between 14.5 and 11.6 ka, though higher reaches on Vintage Peak retained ice until 10-12 ka before retreating to smaller than modern positions. Glaciers on Vintage Peak advanced within 100 m of late Holocene maximum positions around 4-6 ka. Poorly constrained subglacial erosion rates, possible inheritance, and variable mass shielding complicate our ability to more robustly interpret bedrock cosmogenic surface exposure histories. Nine $^{10}$Be ages on late Holocene moraines reveal that glaciers reached their greatest Holocene extents ca. 1300 CE. Our results agree with other regional glacier records and demonstrate the utility of surface exposure dating applied to deglaciated bedrock as a technique to help construct a record of Holocene glacier activity where organic material associated with glacier expansions may be absent or poorly-preserved. Further work to increase exposure/burial history modeling complexity may help to better constrain complex exposure histories in glaciated alpine areas.



## 1 Introduction

Moraines provide the most unequivocal evidence to gauge the magnitude of a glacier advance, however, Holocene moraines deposited prior to the Little Ice Age in North America are rarely preserved (Barclay et al., 2009; Menounos et al., 2009; Solomina et al., 2015). Glacier activity, particularly the timing of glacier advances, prior to the LIA can be elucidated through lateral moraine stratigraphy, overridden wood in glacier forefields, and proglacial lake sediments, among other proxy records (Hawkins et al., 2021; Luckman et al., 2020; Tomkins et al., 2008). Alternative

methods are needed to determine the duration of ice cover at alpine sites in order to place current rates of glacier retreat into a long-term perspective. Cosmogenic surface exposure dating offers a useful method to document past glacier activity (Balco, 2010).

Cosmogenic surface exposure dating measures the concentration of rare isotopes in rock at the surface of the Earth to

calculate the duration the surface has been exposed and/or buried (Balco, 2010; Gosse and Phillips, 2001). For late Pleistocene and Holocene studies, the combination of $^{14}C$ ($t_{1/2}$ 5,730 yr) and $^{10}Be$ ($t_{1/2}$ 1.38 Myr), provide several advantages over the measurement of a single nuclide alone. Due to the relatively short half-life of $^{14}C$, times of previous exposure, such as during Pleistocene interglacials are quickly "forgotten" through radioactive decay. Conversely, $^{10}Be$ decays slowly enough that it is effectively stable over Holocene timescales. In situations of negligible

erosion, $^{10}Be$ acts to record total exposure duration at a site, while $^{14}C$ can elucidate periods of burial. Hippe (2017) provides a clear overview of the concepts behind using paired $^{14}C$ and $^{10}Be$ to investigate Holocene surficial processes. At locations where a well-constrained glacier chronology exists, the cosmogenic nuclide inventory within previously glaciated bedrock surfaces can constrain subglacial erosion rates (Balter-Kennedy et al., 2021; Goehring et al., 2013). Subglacial erosion rates, however, can vary significantly across previously glaciated surfaces (Graham et al., 2023;

Koppes, 2022; Magrani et al., 2022). Past studies of subglacial erosion rates have sampled multiple sites across latitudinal and longitudinal transects of a glacier's flowline to assess the variability of subglacial erosion across zones a differing slope, ice thickness, and resulting ice velocity. The rate of subglacial erosion must be independently estimated in cases where there are few bedrock samples across the formerly glaciated landscape and past glacier activity is unknown or poorly constrained.


In formerly glaciated alpine sites of western Canada, bedrock surfaces that lie beyond limits of late Holocene glacier expansion would have been exposed to cosmogenic nuclide flux since retreat of the Cordilleran Ice Sheet (CIS) following the termination of the Last Glacial Maximum (Menounos et al., 2017, 2009; Seguinot et al., 2016). The CIS, large decaying masses of the CIS, or even cirque glaciers may have experienced cover by re-advance during the

Younger Dryas cold interval, afterwhich the ice sheet retreated rapidly (Menounos et al., 2017). Alpine glaciers in western Canada were likely smaller than their 2000-2010 CE extents for much of the early Holocene (Koch et al., 2014, 2007; Menounos et al., 2004) and may have begun periodically advancing as early as 8 ka, with more evidence for glacier advances of generally increasing magnitude by ca. 6 ka. At most sites in western Canada, glaciers reached their greatest Holocene extent during the Little Ice Age, between 700-150 years before 2000 CE (Menounos et al.,



2009; Mood and Smith, 2015a). What remains unclear is the size of alpine glaciers prior to Neoglacial expansion in the mid-Holocene and the amount of time glaciers were near their late Holocene maximum extents or retreated to within their modern (ca. 2024 CE) extents.

We seek to investigate the record of late Pleistocene through Holocene glacier change at an alpine site in the southern

Coast Mountains of British Columbia, Canada, through several different applications of cosmogenic nuclide surface exposure dating. To determine the duration of ice cover at this site, we make necessary assumptions of the impact of snow cover, subglacial and subaerial erosion, and inheritance on the calculated exposure and burial duration at each bedrock site. This study highlights the utility of paired nuclide dating and bedrock cores to more closely constrain complex exposure histories, the limitations of bedrock exposure dating, and provides new numerical ages for end

moraines in the southern Coast Mountains.

## 2 Study Area

Vintage Peak (1876 m above sea level - asl) is located at the head of Powell Lake on the traditional territory of the Tla'amin Nation in the southern Coast Mountains of British Columbia, Canada (50.25° N, 124.30° W, Fig 1). The mountain is composed of a mid-Cretaceous granodiorite pluton that forms broad sloping surfaces and steep rock walls

that extend down to the valley bottom (Bellefontaine et al., 1994). A small (<0.05 km$^2$) north-facing glacierette herein referred to as "Lockie's Glacier" is located at 1780 m asl in a cirque west of Vintage Peak, below a sub-peak called Lockie's Table. Another north-facing glacierette resides on the north side of Vintage Peak. The cirque headwalls are variably steep, though large portions of the headwall give way to rolling, convex ridgelines (Fig 1).



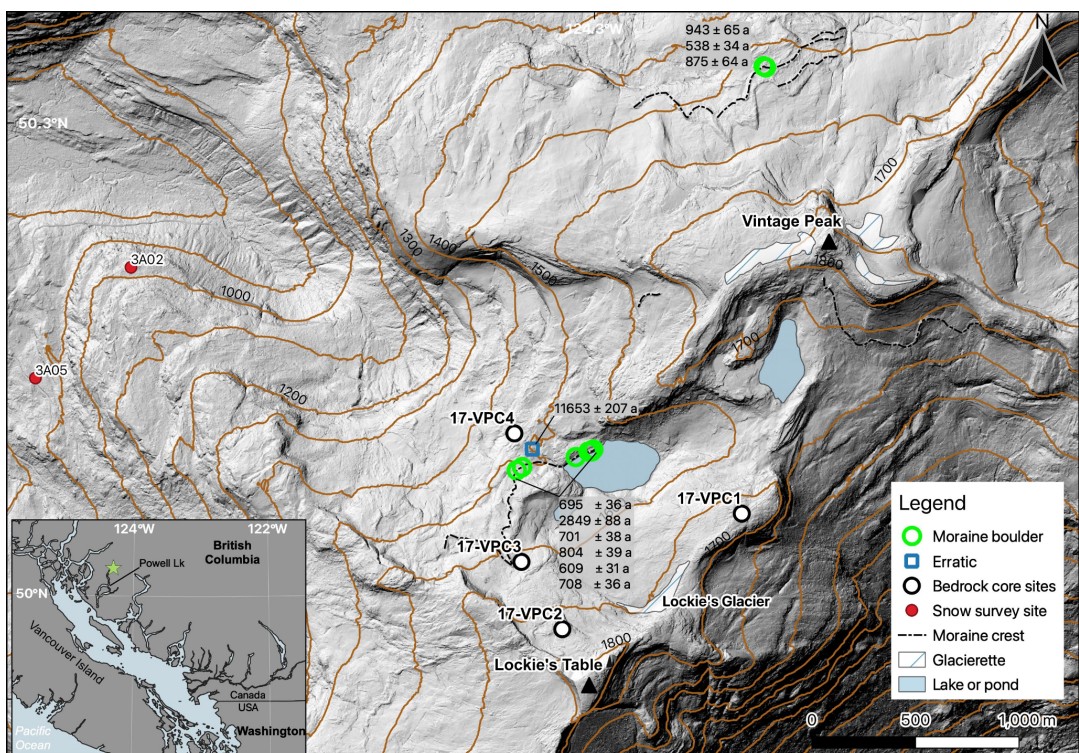

**Figure 1: Location of bedrock and boulder sampling locations on Vintage Peak.** [10]Be surface exposure ages are listed in text boxes with analytical errors. Green star on inset map is the study location. Base hillshade is from a 2017 aerial LiDAR survey. Contour interval is 100 meters.

The study site is 50 kilometers east of the Strait of Georgia, with the coastal climate producing high precipitation and moderate temperatures. Between 1998 and 2022, the mean annual air temperature and mean annual precipitation at Toba Camp (52 m asl), 38 km NE of Vintage Peak was 10.1 °C and 1401 mm, respectively (Env. and Climate Change Canada, 2022). Assuming a lapse rate of 6.5 °C km$^{-1}$, the average annual air temperature at 1600 m on Vintage Peak is close to 0 °C. There are two Province of British Columbia snow survey sites with near annual observations since 1938 CE located just 2 km away from Lockie's Glacier at 910 and 1040 m a.s.l. (Fig 1), that together have a mean annual snow water equivalent (SWE) at the end of March or early April of 847 ± 399 mm water equivalent.

# 3 Methods

## 3.1 Field methods

To constrain the age of the inferred late Holocene moraines in the study area, we sampled nine large boulders on or near the crest of moraines or those resting directly on exposed bedrock. Six boulders represent samples from the



Lockie's Glacier moraine and three originate from the Vintage Peak moraine (Fig 1). In addition, we sampled one erratic (17-VP-07), approximately 60 m beyond the Lockie's Glacier end moraine. At each site, we recorded boulder dimensions, GPS position and elevation, local horizon and boulder self-shielding using a Brunton compass and inclinometer. We used a gas-powered concrete saw and hammer and chisel to collect ~1 kg of rock from each boulder.


A common practice in cosmogenic surface exposure dating on moraines is to sample large boulders from on or near the crest of the moraine, with the aim to reduce the chance of sampling a boulder that has experienced significant snow cover, exhumation, or post-depositional movement (Gosse and Phillips, 2001; Heyman et al., 2016). The Hakai Airborne Coastal Observatory completed an aerial LiDAR and orthoimagery survey of Vintage Peak on September

15, 2017 and again on May 16, 2023. Snow is retained in May on Vintage Peak, while the September survey was largely snow-free. The elevation difference between the co-registered, 1-m spatial resolution digital elevation models (DEMs) gives an approximation of the snow depth in May 2023 across the survey area, notably around the sampled moraines and erratic.

We collected four bedrock core samples within or just beyond the late Holocene extent of Lockie's Glacier (Fig 2). Core sample locations were chosen to capture a variety of potential bedrock exposure histories. At each location, we used a portable drill to collect a pair of 0.35-0.55 m deep, 41 mm outer diameter bedrock cores. Cores were not taken deeper than 0.55 m due to the slow penetration rate of the drill (supplied drill bits for the rental drill were for sedimentary rather than igneous, intrusive rock). A pair of cores within 0.10-0.15 m of each other were taken at each

site to have enough material at each depth for dating. Similar to boulder samples, we recorded sample coordinates, elevation, surface orientation, and horizon shielding at each bedrock site. Samples were transported in plastic tubes, with core orientation labeled and maintained throughout transport and laboratory sampling (Fig 2). Below, we refer to the pairs of cores at each site as a single core.

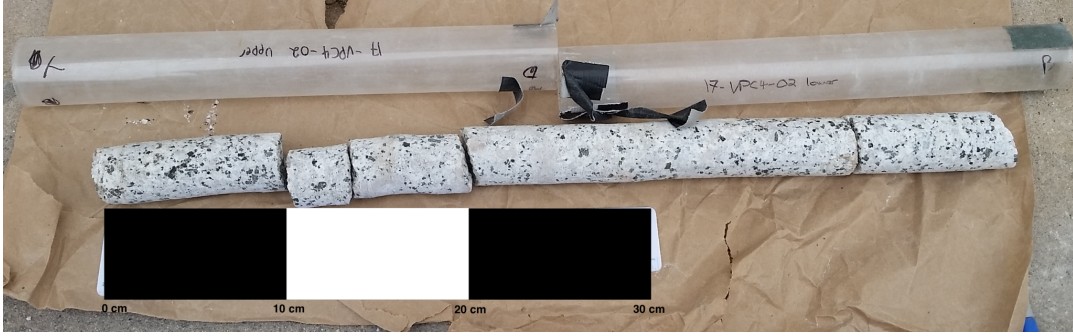

**Figure 2: Bedrock core from site 17-VPC4**. Photo scale converted to digital representation.

One core (17-VPC4) was collected from a bedrock knob, approximately 150 m beyond the presumed late Holocene extent of Lockie's Glacier. Cores 17-VPC3 and 17-VPC2 are within late Holocene glacier limits, with 17-VPC3 ~50

m inside the late Holocene moraine and 17-VPC2 ~40 m beyond the modern (2017 CE) ice limit (Fig 1). Core 17-



VPC1 is located just off a broad ridgeline on the NE side of the Lockie's Table cirque and is presumed to have been covered by minimally erosive ice when Lockie's Glacier was near its maximum Holocene extent. We attempted to sample bedrock with minimal glacier erosion by sampling from areas with oxidized surfaces and preserved striations. We avoided surfaces on the lee of bedrock steps that may be the result of subglacial plucking (Graham et al., 2023; Rand and Goehring, 2019).


### 3.2 Laboratory processing

All samples were processed at the Tulane University Cosmogenic Nuclide Laboratory. We followed the procedure of Nichols and Goehring (2019) to physically and chemically isolate quartz from each sample. Extraction of $^{10}$Be from the sampled boulders and the top 5 cm of each bedrock core followed standard chemical isolation procedures (Ditchburn and Whitehead, 1994). Each batch of approximately eight samples (SM Table 1) included a sample blank.


Each bedrock core was divided into five, 0.05 m-long sections. While both $^{10}$Be and $^{14}$C were measured at the top 0.05 m of each core, we only measured $^{14}$C along the remaining length of each core. We measured both $^{10}$Be and $^{14}$C on the one erratic boulder, 17-VP-07. Extraction of $^{14}$C followed the procedures of Goehring et al. (2019).

Concentrations of $^{10}$Be were measured by accelerator mass spectrometry (AMS) at Purdue University's PRIME Laboratory, with all $^{10}$Be measurements normalized to the 07KNSTD standard (Nishiizumi et al., 2007). We measured $^{14}$C concentrations via AMS at the Woods Hole Oceanographic Institution's National Ocean Sciences Accelerator Mass Spectrometry (NOSAMS) facility. Sample blank corrections for the $^{14}$C samples followed the procedure of Balco et al. (2022), where blank corrections are treated as a lognormal distribution. Exposure ages were calculated using



version 3 of the online exposure age calculator formerly known as CRONUS-Earth (Balco et al., 2008; Borchers et al., 2016). Topographic and self-shielding corrections were applied to all moraine and bedrock sample surfaces. Corrections for snow cover were applied to all bedrock and moraine boulder samples separately and are discussed further below. Moraine ages are presented as a median ± interquartile range. This choice of summary statistics does not rely on the assumption of a normal distribution in our dataset. Median ages and individual sample ages are


presented in years ago (a) before the year of sampling (2017 CE).

### 3.3 Bedrock exposure history modeling

The measured nuclide concentration in each of the bedrock cores represents an integrated signal of exposure, burial by ice and snow, and erosion. While this integrated signal precludes us from being able to constrain the precise timing of Holocene glacier advance and retreat, we can apply reasonable estimates of the subglacial erosion rate and seasonal


snow cover at each site, then determine the most likely cumulative duration of burial and exposure for each site that can be explained by our data.



We assume that Cores 1 and 4 experienced negligible erosion since they were exposed following the termination of the Last Glacial Maximum. Core 1 lies on a broad bedrock shoulder and would have been covered by thin, minimally erosive ice when Lockie's Glacier was at or near its Holocene maximum extent. Core 4 lies outside of the Lockie's Glacier's late Holocene moraines and would have experienced no erosion by ice over the course of the Holocene. Subaerial erosion rates in alpine environments are typically low (Elkadi et al., 2022; Lehmann et al., 2020), rarely more than 0.001 cm yr$^{-1}$. As such, we assume that during periods of exposure, there was negligible subaerial erosion of bedrock surfaces.

Cores 2 and 3 would have been covered by Holocene ice and are expected to have experienced subglacial erosion during periods of burial. Subglacial erosion rates have been shown to be highly variable across regions and within single glacier forefields (i.e. Koppes et al., 2015; Magrani et al., 2022; Rand and Goehring, 2019). While others have imposed a "known" glacier history to model likely subglacial erosion rates (Schimmelpfennig et al., 2022; Steinemann et al., 2021, 2020; Wirsig et al., 2017), we chose to fix a likely subglacial erosion rate in order to estimate Holocene glacier activity. Given our sampling strategy that avoided evidence of late Holocene plucking, subglacial erosion would have primarily been through abrasion. Abrasion rates have been estimated by others to be 0.009-0.035 cm yr$^{-1}$ in the Puget Lowland of Washington State under the Cordilleran Ice sheet (Briner and Swanson, 1998), 0.002-0.033 cm yr$^{-1}$ at the Rhone Glacier in Switzerland (Goehring et al., 2011), and 0.013 ± 0.008 cm yr$^{-1}$ at Jakobshavn Isbrae in Greenland (Graham et al., 2023). As striations on the bedrock at Cores 2 and 3 clearly indicate there was a non-zero amount of erosion of the bedrock surface, we present likely cumulative exposure histories for these sites given a subglacial erosion rate of 0.005, 0.01, and 0.02 cm yr$^{-1}$.

We used the exposure age of an erratic boulder outside of late Holocene glacier limits to constrain the maximum possible exposure duration for all sites, assuming the exposure age of the erratic represents the timing of late Pleistocene deglaciation on Vintage Peak. The minimum exposure duration at each sample location was constrained by the $^{14}$C exposure age of bedrock surface samples. We plotted the measured $^{14}$C/$^{10}$Be ratios on a two-isotope diagram with burial isochrons to determine the apparent burial history of each sample. This is later compared to the model exposure and burial histories discussed below.

All sites are assumed to have been seasonally covered by snow throughout their exposure history. We used the average snow depth and density from nearby snow monitoring sites to apply a snow cover correction for six months of the year. The average (± 1 standard deviation) snow density is 0.38 ± 0.07 g cm$^{-3}$ and average depth is 220 ± 102 cm (SM Table 2). During periods of burial by ice, we assume the ice was sufficiently thick to cease nuclide production.

For each site, we run a Monte Carlo simulation 10,000 times per discrete erosion rate ($\epsilon$) of 0.0, 0.005, 0.01, or 0.02 cm yr$^{-1}$, with each run picking a uniform, randomly distributed value of exposure duration ($t_e$) and burial duration ($t_b$) in years. This results in 40,000 runs per site. Our method is similar to the method presented in Jones et al. (2023), except our model looks at a cumulative burial and exposure history, rather than developing the exposure history in a



stepwise fashion over the Holocene. We chose to make no assumptions about the likelihood of exposure or burial over the course of the Holocene and just limited the possible exposure and burial durations. The exposure duration is a value between the maximum possible exposure duration of 15 ka and the minimum exposure duration, which is the $^{14}$C age of the bedrock surface for each site. The burial duration is a value between zero and the difference between the maximum exposure duration and the minimum exposure duration at each site. The theoretical nuclide

concentration at each depth in the model space was calculated using Eq. 1:

$$N_i = \frac{P_i}{\lambda_i}(1 - e^{(-t_e\lambda_i)})(e^{-\lambda t_b})(e^{\frac{\rho\epsilon t_b}{\Lambda}}),\qquad(1)$$

where $N_i$ is the nuclide concentration (atoms g$^{-1}$) of the isotope in question ($^{14}$C or $^{10}$Be), $P_i$ is the surface production rate (atoms g$^{-1}$ yr$^{-1}$), $\lambda_i$ is the nuclide-specific decay constant (s$^{-1}$), $\rho$ is the rock density (g cm$^{-3}$), and $\Lambda$ is the attenuation length (160 g cm$^{-2}$). We assumed the bedrock has an average density of 2.65 g cm$^{-3}$.


We ran the model setup described above assuming no inherited $^{14}$C or $^{10}$Be. Since inherited $^{10}$Be is more likely, we ran our model again for each site assuming a duration of possible inheritance as indicated by the relative $^{14}$C/$^{10}$Be concentrations at Core 4. This produced another 40,000 total simulations per site.

We tested the influence of a 0.001 cm yr$^{-1}$ subaerial erosion rate on our simulations and found that for Cores 1-3 there was no significant difference in the modeled burial or exposure ages, and at Core 4, inclusion of subaerial erosion lead to no simulations producing exposure and burial histories that matched our observations. Thus, we present our results with no subaerial erosion.

We retained simulations that produced modelled $^{14}$C profiles within the $2\sigma$ error bounds of the measured $^{14}$C profiles and a modeled surface $^{10}$Be concentration within the $2\sigma$ errors of the measured $^{10}$Be concentration. We then calculated the $\chi^2$ statistic to evaluate goodness of fit between the modeled and measured nuclide concentrations with depth. The solution with the lowest $\chi^2$ value was selected as the most likely burial and exposure history for each site.

## 4 Results and Discussion

**4.1 Late Holocene moraine ages**

At their late Holocene maximum, the glacier on the north aspect of Vintage Peak covered a maximum area of nearly 1.9 km$^2$, while Lockie's Glacier covered approximately 1.5 km$^2$. We estimate that glacier ice would have been at least 35-40 m thick when Lockie's Glacier was near its late Holocene maximum extent. This estimate comes from assuming a linear surface slope between the headwall and toe of the glacier of 16°, and a yield stress of 100 kPa (Cuffey and

Paterson 2010). By 2020 CE, both glaciers were less than 0.1 km$^2$ and are now likely too thin to flow.

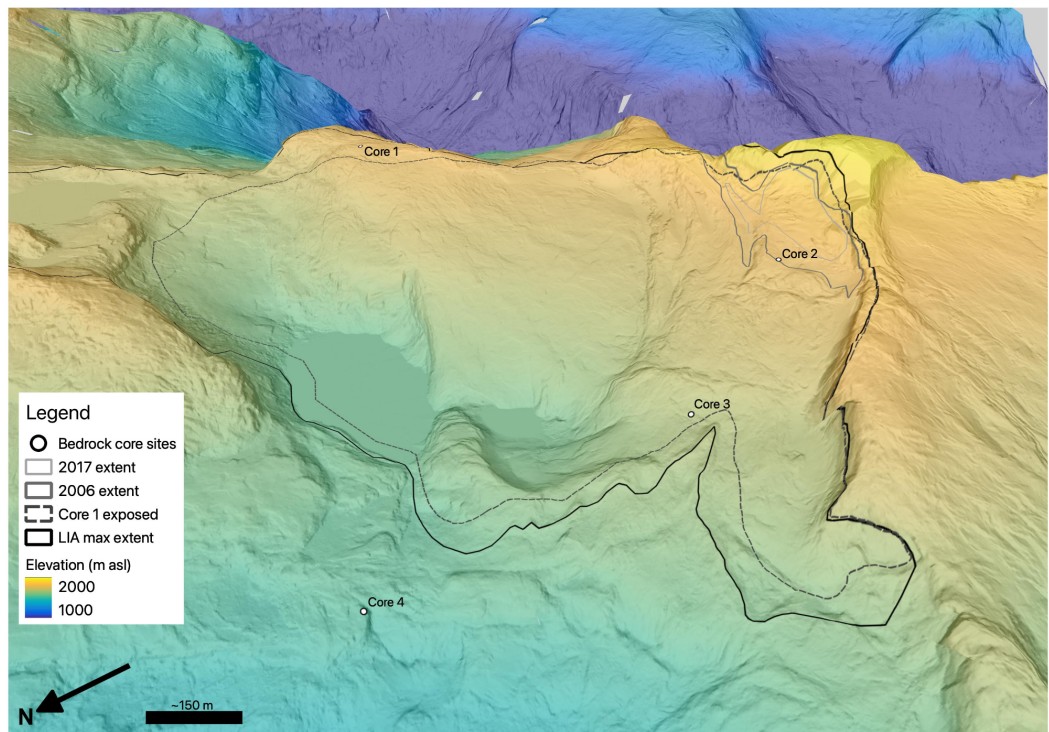

**Figure 3. Lockie's Holocene glacier extent change.** The boundary of Lockie's Glacier when Core 1 was exposed is uncertain but must have at least covered the site of Core 3. Four time periods are shown on this figure: 1) The Little Ice Age maximum extent of Lockie's Glacier; 2) The intermediate extent of Lockie's Glacier that must have covered the site of Core 3, but left Core 1 exposed; 3) the 2006 CE extent; and 4) the 2017 CE extent of Lockie's glacier taken from satellite imagery. Figure basemap is an oblique hillshaded digital elevation model from a 2017 aerial LiDAR survey.

Moraines sampled in this study were bouldery and draped over bedrock, with crests that were no more than 5 m above the bedrock surface (Data S1). Collectively, the nine moraine boulders sampled on Vintage Peak [10]Be date to 705 ± 219 a (1090-1530 CE, Fig 4). Moraine boulders on the Lockie's Glacier and Vintage Glacier moraines date between 934 and 538 a, except a single boulder on the Lockie's Glacier moraine that dates to 2849 ± 88 a.

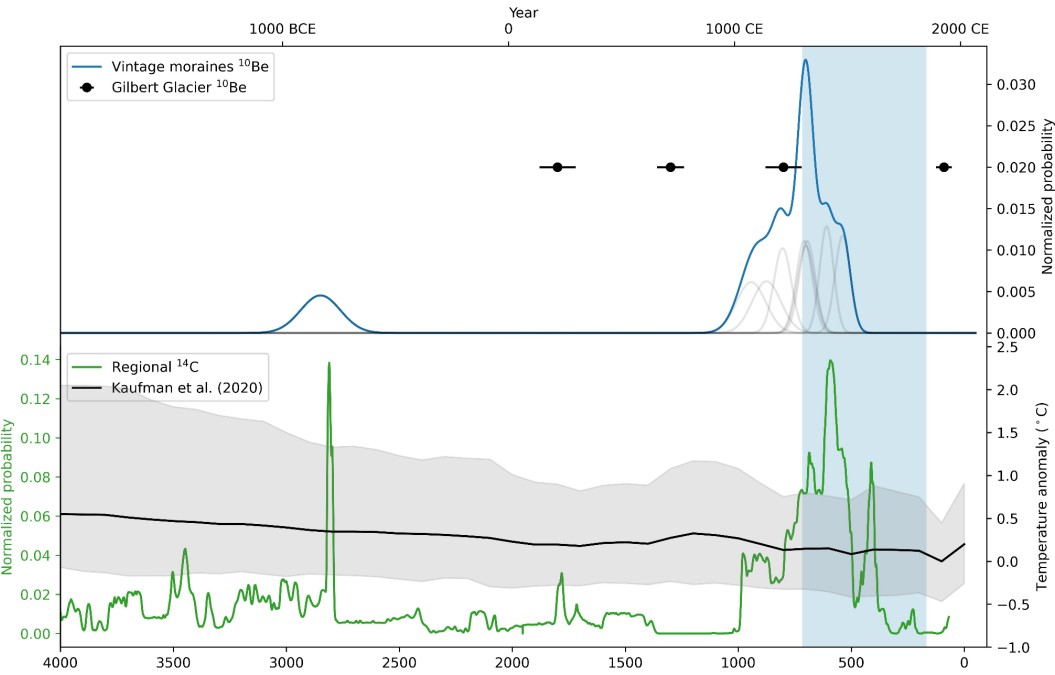

**Figure 4: Top panel: Probability density function of $^{10}$Be moraine ages on Vintage Peak (blue line) with kernel density plot showing individual sample ages.** Black markers are the median ± interquartile range $^{10}$Be ages from moraines fronting Gilbert Glacier, 65 km north of Vintage Peak (Hawkins et al. 2021). Bottom Panel: Normalized probability from organic *in situ* $^{14}$C samples from glacier forefield and composite moraines in western Canada, interpreted to closely constrain periods of glacier advance (modified from Hawkins et al. (2021)). Black line is the mean reconstructed temperature anomaly with respect to the 1800-1900 CE average for the 30 - 60° N latitude band from Kaufman et al. (2020). Blue shaded region is the Little Ice Age period from 1300 - 1850 CE.

Repeat LiDAR surveys completed in the late Spring and Autumn reveal the depth and distribution of snow across Vintage Peak (Fig. 5). In our survey data, moraine crests were covered by up to several meters less snow than adjacent bedrock areas. While annual snow depth will vary, the distribution of snow over the landscape will be broadly consistent each year. Manual snow depth measured at the nearby snow survey sites was ~25% of normal on March 30th, 2023, while an automated snow survey site (3A25P, 1340 m asl) 80 km SE of Vintage Peak similarly recorded the snowpack to be 25% of normal in Mid-May. While the LiDAR surveys support the assumption that sampling boulders for surface exposure dating on moraine crests will reduce the influence of snow cover on moraine exposure ages, we do still apply a snow cover correction to moraine boulders and bedrock as discussed in Methods.



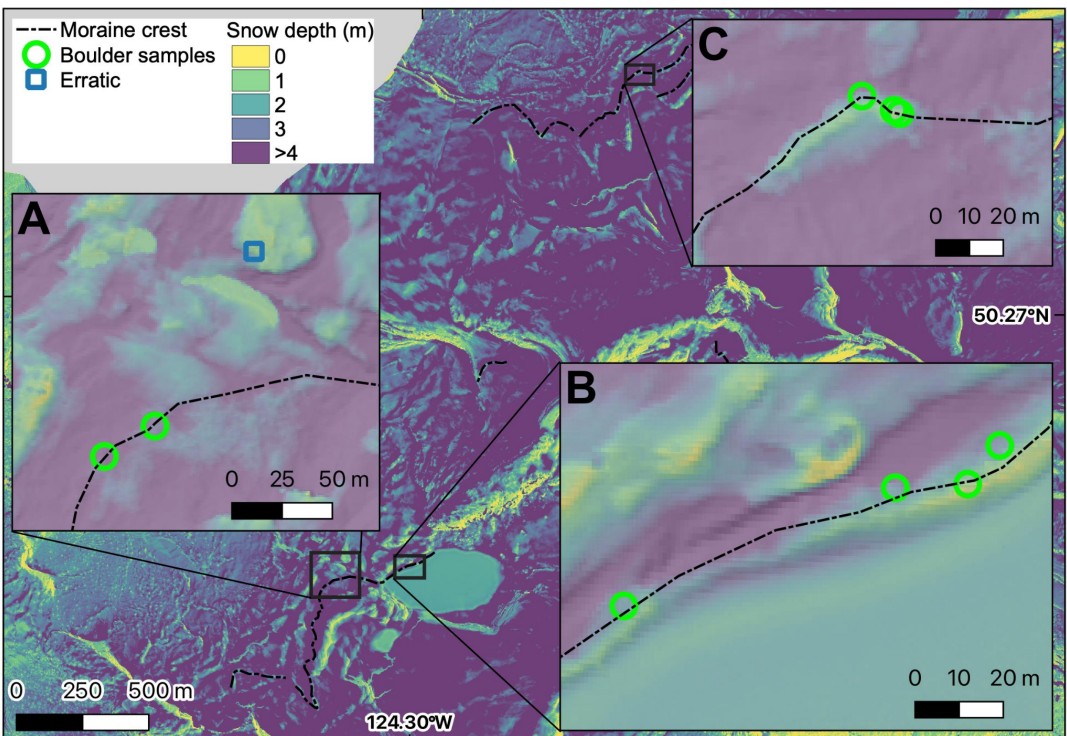

**Figure 5. Snow depth variations on Vintage Peak moraines.** Elevation change between aerial LiDAR-derived DEMs collected May 16, 2023 (snow covered) and September 15, 2017 (snow free). Note the moraine crest within panel A is diffuse, while the moraine is more sharply crested in panels B and C.

Glaciers on Vintage Peak reached their Holocene maxima at the beginning of the classic LIA. The timing of glaciers reaching their late Holocene maximum on Vintage Peak accords with other nearby glacier records (Fig. 4). Nearby glaciers advanced multiple times from 1.4 ka to 0.3 ka (Hawkins et al., 2021; Koehler and Smith, 2011; Mood and Smith, 2015b; Reyes and Clague, 2004; Ryder and Thomson, 1986). The regional radiocarbon record provides evidence of glacier advance, but not the timing of glaciers reaching their maximum positions and depositing moraines. While there is good correspondence among records of glacier advance preceding the age of moraines on Vintage Peak, most glaciers in the region continued to advance after 1300 CE. Though many glaciers in western Canada reached their greatest Holocene extents between 1600 and 1850 CE (Menounos et al., 2009), there is increasing evidence that glaciers were near their late Holocene maximum positions several times as early as 2 ka (Hawkins et al., 2021; Maurer et al., 2012). Stochastic variability in glacier response to climate forcing, or differing response times to climate



perturbations, may explain why glaciers on Vintage Peak reached their greatest Holocene extents slightly earlier than

275     many other glaciers in western Canada (e.g. Roe, 2011).

**4.2 Paired $^{14}$C and $^{10}$Be on erratic and bedrock**

The erratic boulder (17-VP-07), some 65 m beyond the late Holocene moraine that fronts Lockie's Glacier, returned

a $^{10}$Be age of $11.65 \pm 0.2$ ka and a $^{14}$C age of $9.74 \pm 0.9$ ka. When plotted on a paired isotope diagram, the erratic falls

within the field of continuous exposure (Fig. 6). The discrepancy between the apparent $^{10}$Be age and $^{14}$C age could be

280     the result of inheritance. The erratic may contain the equivalent of ~2 kyr of inherited $^{10}$Be, which is more likely than

the boulder having any significant quantity of inherited $^{14}$C. Conversely, another way to produce the observed

difference in $^{10}$Be and $^{14}$C ages on the erratic is by mass shielding under seasonal snow cover. Mass shielding, in this

case by snow, increases the calculated age of a surface. $^{14}$C is more sensitive to thin mass shielding, which means that

the apparent $^{14}$C age increases faster than the apparent $^{10}$Be age with increasing shielding correction (Hippe, 2017).

285     With a shielding factor that would be produced by 190 cm of 0.38 g cm$^{-3}$ snow for six  months of the year, both

nuclides yield an equivalent exposure age of ~14.5 ka. Given the average snow depth of $220 \pm 100$ cm and average

density of $0.38 \pm 0.67$ g cm$^{-3}$ (n=123, mean ± standard deviation) at nearby snow monitoring sites which are nearly

600 m a.s.l. lower than 17-VP-07, this snow cover value is reasonable. Work by Menounos et al. (2017) and Seguinot

et al. (2016) provide cosmogenic ages and ice sheet modeling that suggest the area around Vintage Peak was likely

deglaciated around 14 ka, giving further confidence that the snow-corrected age of the erratic is an accurate

deglaciation age for Vintage Peak. However, as we cannot rule out the possibility of inheritance, we constrain the age

of deglaciation on Vintage Peak to 14.5 ka - 11.6 ka.



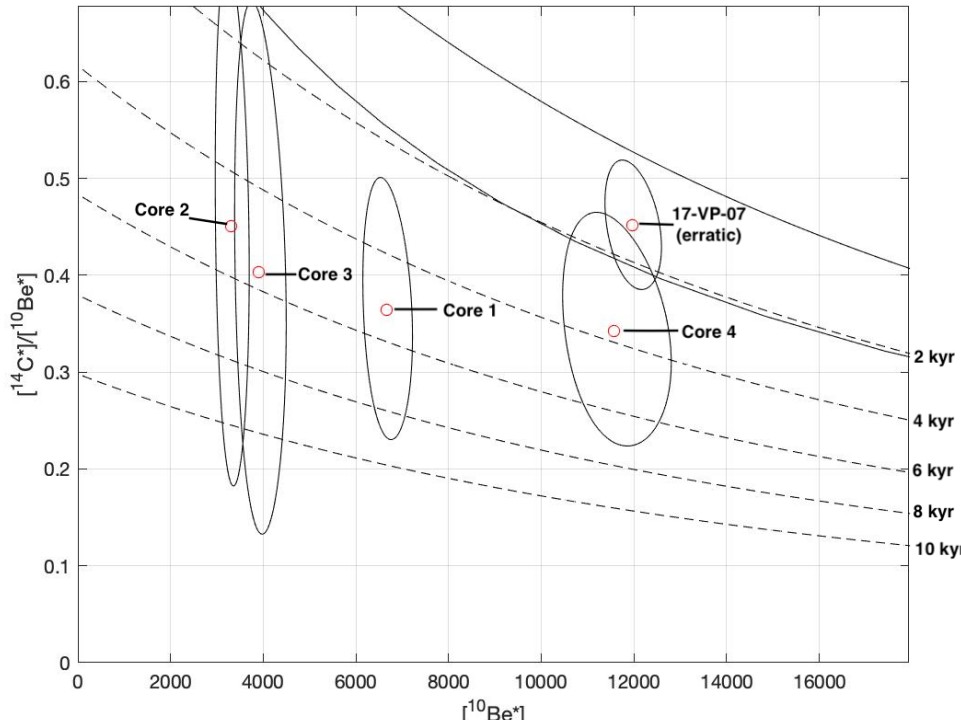

**Figure 6: Paired $^{14}$C/$^{10}$Be nuclide diagram with y-axis normalized to the production ratio of $^{14}$C/$^{10}$Be and x-axis (atoms/g $^{10}$Be) normalized to the site-specific production ratio of $^{10}$Be.** Dashed lines represent burial isochrons, the uppermost indicating 2 kyr burial, with each descending line representing an additional 2 kyr of burial. The uppermost black solid line is the line of continuous exposure with zero erosion, and lower black line is the line of steady-state equilibrium; samples plotting between these two lines were continuously exposed and eroding at a given rate.

The erratic boulder and lack of apparent Younger Dryas moraine(s) indicates that glaciers on Vintage Peak did not extend beyond their late Holocene positions since late Pleistocene deglaciation that may have commenced as early as 14.5 ka. Some cirque glaciers in western Canada that were above the decaying CIS advanced during the Younger Dryas deposited moraines on the landscape (Menounos et al., 2017). However, we did not observe any moraines beyond the late Holocene moraines at Vintage Peak. Our findings on Vintage Peak further support that, on average, glacier advances during the Little Ice Age were the most extensive since the termination of the LGM.

Both $^{10}$Be and $^{14}$C bedrock surface concentrations at each core site follow expected trends based on geomorphic position relative to the modern glacierette (Fig. 7). The greatest nuclide concentrations are found at Core 4 (17-VPC4), outside of the late Holocene moraines, with a surface $^{10}$Be concentration of 1.669 ± 0.0167 x 10$^5$ atm g$^{-1}$ and $^{14}$C




concentration of $1.886 \pm 0.445 \times 10^5$ atm g$^{-1}$. Nuclide concentrations at Core 3 (17-VPC3, just inside of the late

Holocene moraine) and Core 2 (17-VPC2, just distal from the 2017 CE glacier terminus) are equivalent within errors

(Table 1). This suggests that, within our measurement uncertainty, the bedrock surface at Cores 2 and 3 experienced

nearly equivalent exposure histories over the Holocene. Cores 2 and 3 have 63-66% less $^{10}$Be and 55-57% less $^{14}$C

than Core 4. We interpret equivalent nuclide concentrations at Core 2 and 3 to indicate that Lockie's Glacier responded

to climate perturbations in a roughly binary manner; existing near the glacier's late Holocene maximum or retracted

close to modern extents.

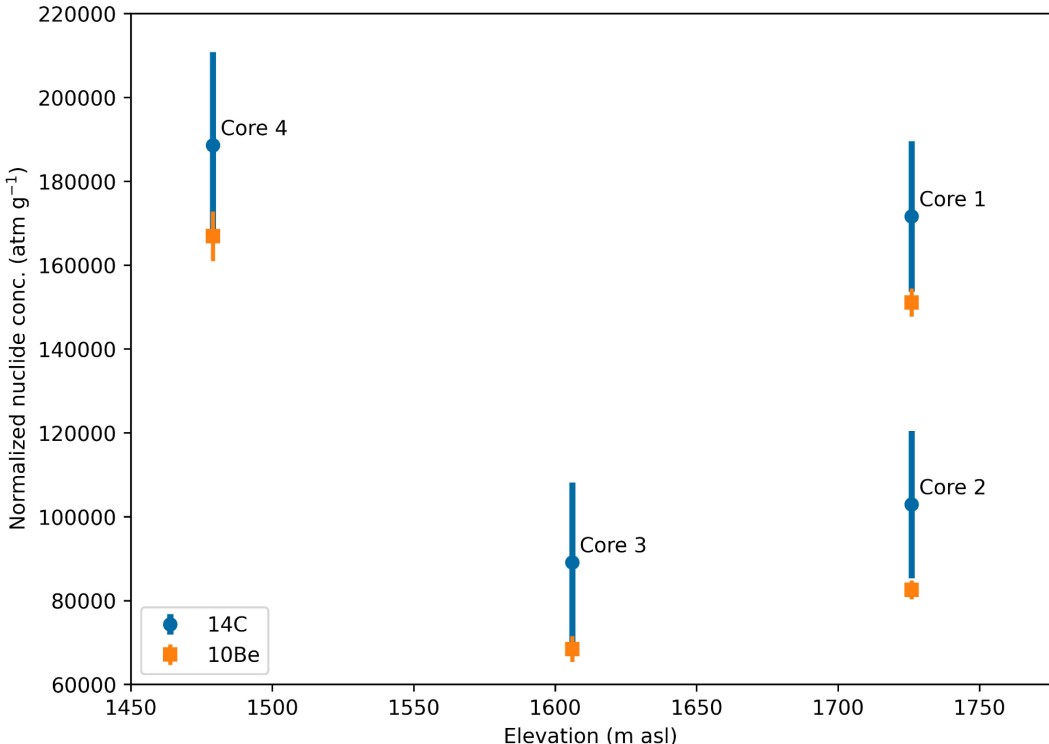

**Figure 7: $^{10}$Be and $^{14}$C concentrations in bedrock cores with elevation.** Nuclide concentration is normalized relative to Core 4, which had the lowest production rate of the sample locations.

The $^{10}$Be and $^{14}$C surface concentrations of Core 1 (17-VPC1), which is assumed to have experienced variable cover

by minimally erosive ice, but less total ice cover than Core 2 or 3, are $1.179 \pm 0.0351 \times 10^5$ atm g$^{-1}$ and $1.401 \pm 0.1795$

$\times 10^5$ atm g$^{-1}$, respectively. Core 1 has 29% less $^{10}$Be and 26% less $^{14}$C than Core 4.




Vintage Peak 10Be Sample information

| Sample | Latitude | Longitude | Elevation | Thickness | Shielding | Quartz (g) | Carrier added | 10Be/9Be ratio | 1 sigma | Blank-corrected | Blank-corrected | Exposure | Exposure | Exposure age | Exposure age |
|---|---|---|---|---|---|---|---|---|---|---|---|---|---|---|---|
| *Lockie's Glacier moraine* | | | | | | | | | | | | | | | |
| 17-VP-01 | 50.26128 | -124.29956 | 1526 | 2.5 | 0.9891 | 49.5115 | 0.2589 | 3.22E-14 | 1.57E-15 | 1.15E+04 | 5.88E+02 | 695 | 36 | 872 | 45 |
| 17-VP-02 | 50.26120 | -124.29966 | 1512 | 2.5 | 0.9889 | 50.0079 | 0.2593 | 1.17E-13 | 3.39E-15 | 4.19E+04 | 1.30E+03 | 2849 | 88 | 3608 | 112 |
| 17-VP-03 | 50.26119 | -124.29989 | 1513 | 2.5 | 0.9891 | 49.9900 | 0.2592 | 3.25E-14 | 1.66E-15 | 1.15E+04 | 6.13E+02 | 701 | 38 | 881 | 47 |
| 17-VP-04 | 50.26096 | -124.30090 | 1512 | 2.5 | 0.9891 | 50.0071 | 0.2596 | 3.71E-14 | 1.72E-15 | 1.31E+04 | 6.39E+02 | 804 | 39 | 1029 | 50 |
| 17-VP-05 | 50.26046 | -124.30437 | 1502 | 2.5 | 0.9889 | 50.0104 | 0.2600 | 2.77E-14 | 1.34E-15 | 9.76E+03 | 4.99E+02 | 609 | 31 | 756 | 39 |
| 17-VP-06 | 50.26032 | -124.30472 | 1504 | 2.5 | 0.9882 | 50.0020 | 0.2896 | 2.92E-14 | 1.40E-15 | 1.19E+04 | 5.83E+02 | 708 | 36 | 891 | 45 |
| *Erratic boulder* | | | | | | | | | | | | | | | |
| 17-VP-07 | 50.26125 | -124.30370 | 1496 | 2.5 | 0.8624 | 50.0114 | 0.2910 | 3.84E-13 | 5.58E-15 | 1.60E+05 | 2.77E+03 | 11653 | 207 | 14853 | 264 |
| *Vintage Glacier moraine* | | | | | | | | | | | | | | | |
| 17-VP-09 | 50.27804 | -124.28846 | 1525 | 2.5 | 0.9473 | 46.7245 | 0.2870 | 3.53E-14 | 2.37E-15 | 1.53E+04 | 1.03E+03 | 943 | 65 | 1251 | 87 |
| 17-VP-10 | 50.27798 | -124.28831 | 1518 | 2.5 | 0.9964 | 50.0095 | 0.2894 | 2.25E-14 | 1.34E-15 | 9.08E+03 | 5.53E+02 | 538 | 34 | 669 | 42 |
| 17-VP-11 | 50.27801 | -124.28834 | 1527 | 2.5 | 0.9964 | 50.0186 | 0.2887 | 3.69E-14 | 2.63E-15 | 1.50E+04 | 1.07E+03 | 875 | 64 | 1149 | 84 |
| *Bedrock coretop/surface* | | | | | | | | | | | | | | | |
| 17-VPC1 | 50.25861 | -124.28943 | 1726 | 5 | 0.9996 | 17.5603 | 0.2593 | 1.16E-13 | 3.05E-15 | 1.18E+05 | 3.35E+03 | 6489 | 185 | 7974 | 227 |
| 17-VPC2 | 50.25346 | -124.30145 | 1726 | 5 | 0.9717 | 18.7625 | 0.2600 | 5.98E-14 | 2.24E-15 | 5.69E+04 | 2.24E+03 | 3376 | 133 | 4196 | 165 |
| 17-VPC3 | 50.25635 | -124.30430 | 1606 | 5 | 0.9739 | 15.7779 | 0.2590 | 5.42E-14 | 2.65E-15 | 6.10E+04 | 3.09E+03 | 3937 | 199 | 4898 | 248 |
| 17-VPC4 | 50.26192 | -124.30493 | 1479 | 5 | 0.9951 | 11.1266 | 0.2608 | 1.03E-13 | 3.48E-15 | 1.67E+05 | 5.93E+03 | 11031 | 393 | 14102 | 502 |

Ages are calculated using version 3 of the online exposure age calculator formerly known as the CRONUS-Earth online exposure age calculator found at https://hess.ess.washington.edu/ (wrapper 3.0.2, muons: 1A,
Be carrier 2014.10.20 used for all samples and contains as Be concentration of 1014 ppm
All Be concentration were measured at the Purdue Rare Isotope Measurement Laboratory (PRIME). Be-10/Be-9 ratios are not corrected for Be-10 detected in procedural blanks.
Shielding by snow cover applied to bedrock core surfaces assuming 220 cm of 0.38 g/cm3 snow cover for 6 months of the year.

**Table 1: Vintage Peak $^{10}$Be sample information.**

### 325   4.3 $^{14}$C bedrock depth profiles

Five measurements of $^{14}$C concentration along the depth of each core show decreasing or consistent, within uncertainties, nuclide concentrations. At Core 4, the $^{14}$C concentration is $1.886 \pm 0.445 \times 10^5$ atm g$^{-1}$ in the upper 0.05 m of the core, and decreases to $1.464 \pm 0.225 \times 10^5$ atm g$^{-1}$ at the bottom 0.40-0.45 m of core (Fig 8, SM Table 3). There is no clear pattern to measurement uncertainty with depth in each of the four cores, with measurement
uncertainty ranging from 10% to 47% (average 25%) of the mean concentration.

Cores 2 and 3 have quasi-vertical $^{14}$C concentration profiles of equivalent magnitude, within errors (Fig 8). Core 1 has a slight decrease in $^{14}$C concentration in the upper section of core, but within uncertainties is quasi-vertical as well.



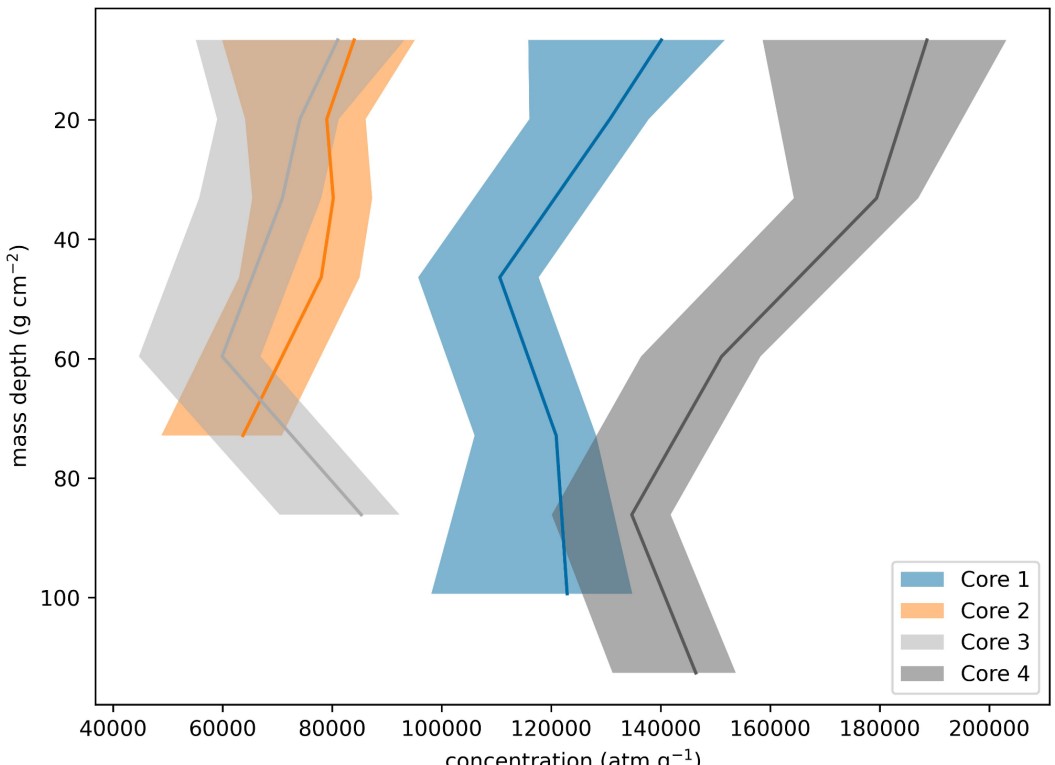

**Figure 8: Bedrock ¹⁴C profiles with depth.** Solid lines are the mean $^{14}$C concentration, error bounds are 1-sigma measurement uncertainties at each sample depth.

## 4.4 Monte Carlo simulations

Monte Carlo simulations of possible burial and exposure histories provide a range of solutions that adequately explain our data. Subglacial erosion and possible inheritance from previous bedrock exposure significantly influences the apparent exposure age of the bedrock. Though we are unable to independently determine the magnitude of inheritance or subglacial erosion at each site, we make reasonable estimates of these values given each site's geomorphic position.

Core 1, assuming no subglacial erosion, experienced 9.3 ka of exposure and 4.9 ka of burial. Assuming complete resetting of the cosmogenic nuclide inventory during the LGM, this means that Core 1 became ice-free following the LGM around 14.3 ka.



Cores 2 and 3 likely experienced some amount of subglacial erosion during the Holocene, as evidenced by striations on the sampled bedrock surface. However, if we assume negligible erosion, Cores 2 and 3 experienced 4.5 and 5.4 ka of exposure and 5.5 and 5.9 ka of burial, respectively. If we invoke a subglacial erosion rate of 0.005 cm yr$^{-1}$, then Core 2 would have experienced 6.4 ka of exposure and 4.5 ka of burial, while Core 3 experienced 7.6 ka of exposure

and 4.6 ka of burial. At both sites, increasing the subglacial erosion rate notably increases the modelled exposure duration and decreases the burial duration. With a subglacial erosion rate of 0.01 cm yr$^{-1}$, the measured nuclide concentration at Core 2 is best explained by 8.5 ka exposure and 3.8 ka of burial, while Core 3 potentially experienced 9.9 ka of exposure and 3.8 ka of burial. Core 4 would have experienced no subglacial erosion since deglaciation following the LGM. Our nuclide measurements at Core 4 are best explained by 14.9 ka of exposure and 2.9 ka of

burial.

While the apparent exposure age of Core 4 is coincident with the timing of regional deglaciation, the nearly 3 ka of burial is at odds with our assumption that Core 4 has experienced continuous exposure following deglaciation. The apparent burial signal may be the result of inheritance due to insufficient resetting of the nuclide inventory in the bedrock during the LGM, mass shielding from transient till cover following deglaciation and/or more snow cover than

we have accounted for, or a combination of these factors. Samples from bedrock sites with a comparable geomorphic context as Core 4 would help indicate if bedrock in this area was fully reset during the LGM, or if areas that did not experience ongoing Holocene glaciation contain inherited nuclides.

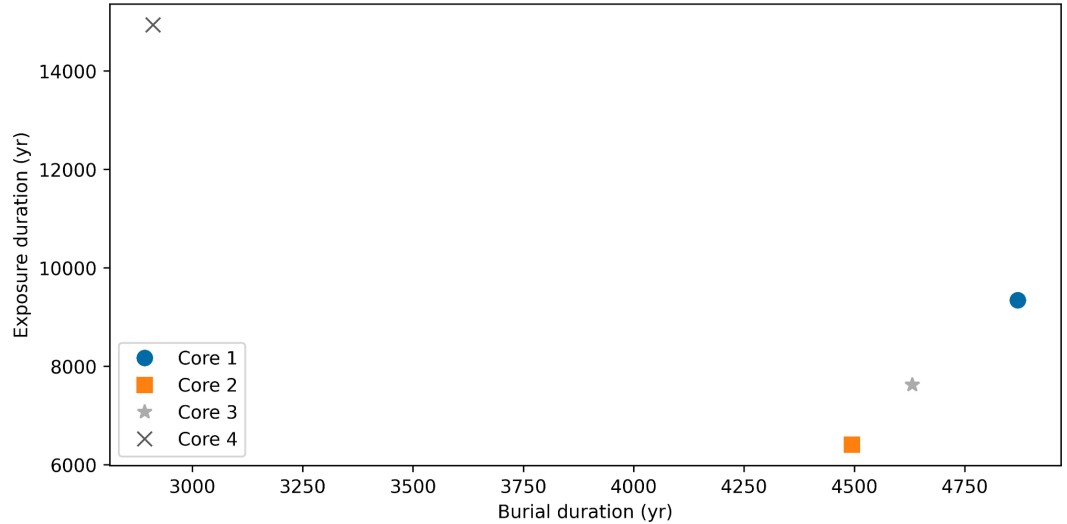



**Figure 9. Summary of Monte Carlo exposure/burial durations.** The most likely subglacial erosion rates for each

core are described in text.

In summary, Monte Carlo simulations provide insight to the effect of varying subglacial erosion on the cumulative

burial and exposure histories that can be explained by our data. At Cores 2 and 3, a small increase in subglacial erosion

would have to be explained by greater total exposure duration and less burial by ice. Preserved striations on the bedrock

surfaces of Cores 2 and 3 demonstrate that some amount of subglacial erosion did occur. A subglacial erosion rate of

0.005 cm yr$^{-1}$ indicates that ice may have covered Cores 2 and 3 until 10.9-12.2 ka. Core 4 records 14.9 ka of exposure

but contains an apparent burial signal of nearly 3 ka that can be explained by inheritance, transient mass shielding, or

a combination of the two.

### 4.5 Results synthesis and regional context

Based on our results and comparison to published glacier records from the region, we conclude that Vintage Peak was

fully covered by erosive Cordilleran Ice Sheet during the Last Glacial Maximum. As early as ~14.5 ka, the Cordilleran

Ice Sheet had wasted enough to expose most of the bedrock surface around Vintage Peak, though alpine glaciation

kept bedrock within the late Holocene glacier maxima covered. Following deglaciation, the site of Core 4 may have

been covered by till for ~3 kyr before the till was eroded away. While we cannot rule out the possibility of former till

cover, we do not observe preserved late Pleistocene till in nearby proglacial environments, which suggests this

explanation is unlikely. Over the course of the Holocene, Lockie's Glacier was larger than present for 4.5-4.6 ka and

smaller than present for 6.4 ka. Glaciers on Vintage Peak reached their greatest Holocene extent 700 ± 220 a (1090-

1530 CE) and have since retreated to minimal extents.

Decreasing solar insolation in the northern Hemisphere strongly influenced the growth of glaciers in western Canada

from the mid-Holocene until the culmination of the Little Ice Age (Menounos et al., 2009; Solomina et al., 2015).

Holocene temperature reconstruction by Kaufman et al. (2020) shows a peak in Holocene temperatures at around 6.8

ka, followed by a consistent decrease in temperature until around 1850 CE (Fig. 4). While our bedrock exposure data

does not allow us to determine distinct times of glacier advance or retreat, regional temperature reconstructions and

nearby glacier records suggest Lockie's Glacier likely advanced to position near its late Holocene maximum extent

during the last 6 kyr.



Our results are at odds with the interpretations of other surface exposure dating studies, namely Menounos et al. (2017) and Darvill et al. (2022). Without correcting for snow cover, the $^{10}$Be ages for the bedrock surface at Core 4 and the erratic agree with the median age of cirque moraines across western Canada formed in response to Younger Dryas cooling (Menounos et al., 2017). However, our paired $^{14}$C-$^{10}$Be dating of Core 4 and the erratic reveal a more complex
history of exposure that is best explained by significant influence by snow and in the case of Core 4, inheritance and/or transient mass shielding following deglaciation. Measurements of $^{14}$C on boulders presented in Menounos et al. (2017) would show whether the boulders have experienced a complex exposure history.

## 5. Conclusion

Our study used paired $^{14}$C-$^{10}$Be dating on alpine bedrock surfaces and bedrock cores to elucidate glacier activity prior
to the late Holocene maximum extent of an alpine glacier in the southern Coast Mountains of British Columbia. We estimate late Pleistocene deglaciation at Vintage Peak to have occurred around 14.5 ka. Bedrock outside of Holocene ice extents suggests limited burial that cannot be explained by snow cover alone and may be the result of previous cover by till or inherited $^{10}$Be. A broad ridgeline that would have been covered by minimally erosive ice during maximum ice extents indicates around 9.3 ka of exposure and ~4.9 ka of burial since late Pleistocene glaciation.
Bedrock exposure histories close to the late Holocene maximum extent and the much less extensive 2017 CE ice margin are nearly equivalent and indicate that the bedrock at this site may not have become exposed following late Pleistocene deglaciation until 10-12 ka. Ice was more extensive than the 2017 CE glacier extent for around 4.5-5.9 kyr. Glaciers on Vintage Peak were likely episodically larger than present from the Neoglaciation until the culmination of the LIA, in response to cooling beginning in the mid-Holocene. Nine $^{10}$Be ages from boulders on two moraines that
front Vintage Peak Glacier yield a median exposure age of 700 ± 220 a (1090-1530 CE), dating the late Holocene maximum extent of glaciers on Vintage Peak to the earlier portion of the Little Ice Age. Future work to test additional simulated exposure histories to investigate the impact of transient till cover and variable pre-LGM inheritance could further constrain glacier change throughout the Holocene. Bedrock exposure ages are influenced by several variables (exposure, burial, inheritance, mass shielding, and erosion); confident determinations of actual exposure and burial
durations require independent constraints on several of these variables.



*Author Contributions.*

Following CRediT Authorship Guidelines, AH contributed to all authorship components except funding acquisition,
administration, resources, supervision, and validation. BG contributed to all components except funding acquisition,
investigation, and original draft. BM contributed to all components except data curation, formal analysis, software,
validation, visualization, and original draft.

*Competing Interests.*

The authors declare that they have no conflict of interest.

*Acknowledgements.*

This research was funded by the Hakai Institute (Tula Foundation), Natural Sciences and Engineering Research
Council (NSERC) of Canada (B.M.) and the Canada Research Chairs Program (B.M.). Jane Markin assisted with field
work. We thank Grizzly Helicopters for their support accessing our remote field site. McElhanney provided a last-
minute rental of their rock drill, crucial to the success of this research. We are grateful to have been able to conduct
our research on the traditional territories of the Klahoose and Tla'amin First Nations.



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
