# Peer review of "Terrestrial Cosmogenic Nuclide depth profiles used to infer changes in Holocene glacier cover, Vintage Peak, Southern Coast Mountains, British Columbia"

_EGUsphere, 2024_

## Author Response (AR1)

RC1:

My general comments:

This paper presents TCN-based geochronology, near Vintage Peak in the southern Coast Mountains of British Columbia, Canada, using nine moraine boulder samples, one erratic block, and four shallow bedrock cores. The main purpose of the study was to reveal the early-to-mid Holocene exposure/burial history of the sampled sites using a multiple-nuclide approach ($^{14}C$, $^{10}Be$) and bedrock depth profiles. However, the complex history of ice, possible inherited nuclide inventory, and likely snow/ice/till cover make the study's outcome somewhat ambiguous. However, authors tried to resolve the glacier history at Vintage Peak, and found out (i) major ice sheet deglaciation roughly between 14.5 and 11.6 ka, ii) Holocene advancement at around 4-6 ka, and iii) further advance of ice to maximum position at the LIA (ca 1300 CE). The paper is well-written, concise, and provides all the necessary details for the community. It certainly deserves publication in the Geochronology journal but requires some major revisions, as suggested by my specific comments.

**We thank the referee for taking time to review our paper and providing useful feedback on how to improve it.**

Specific comments:

• Explaining the complex history of alpine ice is a challenging task, but the authors should address and deal with specific complexities individually, such as till cover, inheritance, bedrock erosion, etc. to show the outcome of the study is valid. For instance, till coverage on bedrock surfaces has not been discussed, and may have influence on accumulation of cosmogenic nuclides depending of the thickness and density of till on top of sampled bedrock profiles.

**A fair point. While we mention the potential impact from transient till coverage on line 359 and lines 379-381, we recognize this discussion is overly brief. Our revised manuscript will discuss this effect in greater detail.**

• The Monte Carlo approach has not been discussed in detail. Some information is presented in Ch. 4.4 and Figure 9, but it's method and results are unclear.

**Our revised manuscript will provide additional details on our Monte Carlo approach. We will also include additional information about the results in the main section of the paper.**

• The connection between the LIA extent and Holocene glaciers is poorly explained. The spatial extents should be elaborated, on maps and in the text.

**We will review and clarify how we explain the relative extent of glaciers on Vintage Peak and will revise Figure 3 and associated text to make our interpretations of relative ice cover history more clear.**

• Cosmogenic $^{14}C$ bedrock depth profiles have not been discussed in sufficient detail.

We agree our current text was too brief when discussing the 14C bedrock depth profiles, their interpretation, and their utility to this study. We will provide additional detail on the first-order observations from the depth profiles and the usefulness for further constraining possible bedrock exposure histories beyond what a surface sample can provide alone.

Technical corrections:
• Title: Since the depth profiles were taken from the bedrock, you may want to add word "bedrock" after the Nuclide, but before the depth, so that the first line of the title will read "Terrestrial Cosmogenic Nuclide bedrock depth profiles used to infer…" There are not only depth profiles in the study, 9+1 boulders, so, the current title is somewhat misleading.

This is a valid point and we will alter our title to include bedrock and address that we sampled multiple types of surfaces for cosmogenic nuclide analysis, to develop our glacier chronology.

• L93: The Vintage peak is located at 1800 m. Its impressions look like it is located at 1600 m.

Figure 1 includes labeled contours showing both Lockie's Table and Vintage Peak are just over 1800 m in elevation. In Line 93, we estimate the average annual temperature at 1600 m, as this is an approximate average elevation from which our bedrock and moraine boulders were sampled. We think Figure 1 and text are clear as written and, unless we are mistaken, we propose to leave the text and figure unchanged.

• L95: On the Table 2, where the snow survey data is presented, the altitude of snow stations are different than the numbers here.

Thank you for this correction. In 2015, the elevation of both snow survey sites was changed. We had listed the older elevation in text, not the most recent elevations as reported by the Province of British Columbia. We will note this change in reported elevation in text.

• L140: SM Table 1 is related to blank samples, not batch of 10Be samples, so move the referring (SM Table 1) to the end of the sentence

Thank you, we will move reference to SM Table 1 to the end of the sentence.

• The erratic boulder (17-VP-07), how would you sure that this is an erratic boulder? a rockfall from top?

This is a fair point. We cannot definitively rule out that 17-VP-07 may be from a rockfall event (albeit an old one). The boulder is subangular and we did not observe any distinctive evidence of glacial transport. However, if this was a rockfall deposit, we would expect to find several other similar clasts in the area, but there was just one smaller

**boulder 10's of meters away. We will provide additional detail on the geomorphic position of 17-VP-07 in our revised manuscript.**

• L323: Most of the columns of this table (Table 1) has no units and last four columns of Table 1 have no explanations.

**We apologize for this formatting error (conversion of the Google Document into a PDF causes several lines to be dropped from the table).**

• Table 3: It is not clear what are the numbers at the last 3 columns of Table 3?

**This is also due to the formatting error mentioned above and will be fixed in the revised manuscript.**

RC2:
Overview:

The manuscript describes the utility and application of Terrestrial Cosmogenic Nuclides Beryllium-10 and *in situ* Carbon-14 (C-14) from boulder, bedrock surface, and bedrock depth profiles from cores to quantify and model glacier exposure/burial scenarios during the Holocene at Vintage Peak, Southern Coast Mountains, British Columbia. At the time of sample collection in 2017, very few, if any, published studies of portable backpack drilling for cosmogenic depth profile analysis had been done, so the application is a novel technique. Modelling for exposure/burial/erosion scenarios were leveraged to characterize glacier size in concert with modern observations of glacier extent and snow depth partial shielding. Results show uncovering from the Cordilleran Ice Sheet around 11.6-14.5 ka, and Vintage peak retaining local ice until 10-12 ka before retreating smaller than modern limits. The ice then began to regrow during the late Holocene/Little Ice Age, and then retreated to modern limits.

The technique of bedrock depth profile shows promise, and C-14 has applications for complex exposure/burial modelling due to the short half life and small mass of quartz necessary compared to Be-10. Difficulties observed during the study were limits of drill penetration due to bit type (this was a common problem with the early portable drill systems cutting bits designed for sedimentary rock) that prevented recovery of the deeper muon dominant depth section, and the relatively larger uncertainty with C-14 compared to Be-10.

The study is generally well written with occasional grammatical inconsistencies or regional word spelling variation. It is in my opinion that the geochronologic techniques applied, in particular the application of *in situ* C-14 in bedrock depth profiles, is a novel and powerful technique that can and should be applied in more studies. This study tested the method and experienced some difficulties but exhibits the utility and need for additional efforts in furthering *in situ* C-14 and depth profiles. There are additional analyses that could have been done regarding C-14 that are

currently in the literature but were not discussed as part of this study and may provide merit, but the authors state there is future work that could be applied to better understand the data. The scientific quality is good, and additional analyses would improve the study, but the results and techniques presented are appropriate for the conclusions made. There is a concern about inheritance and incomplete nuclide resetting that may be addressable through additional evaluation of C-14 modelling from Secular equilibrium (ex Schweinsberg et al,2018; Graham et al, 2019; Sbarra et al, 2022) or consideration of deep muon long term production (Briner et al, 2016; Balter-Kennedy et al, 2021). The presentation quality is good, and minor revision would improve the quality of the manuscript.

**Thank you for your time and comments that will improve our manuscript.**

General notes:

There are some parts of the manuscript that could be improved. In general, the application mindset of the utility of cosmogenic nuclide analysis appears to be strictly focusing on the method as a chronology technique, however, studies referenced in the paper expand on using the method for more than just chronology (ex Balter-Kennedy et al, 2021, Graham et al, 2023). Part of the utility of depth profile analysis and modelling is the *a priori* knowledge of the glacial history to constrain the modelling parameters. When referring to the depth profile, it may be more accurate to refer to it as "cosmogenic nuclide analysis" or "analysis" rather than "dating" because the methods can infer more than an age. Alternatively, the authors could refer to the measured concentrations as "relative exposure ages" for comparison of shielded samples that are lower than the expected age if the sample was not affect by additional mass depth (example Graham et al, 2023). "Relative exposure ages" can be useful in relating a more audience understandable metric of difference.

**We intentionally limit *a priori* knowledge of past glacier activity in our depth profile analysis. Unlike in the studies mentioned above, we have limited direct evidence of glacier cover through the Holocene at Vintage peak, and the relative paucity of bedrock samples from this site lead to us deciding to focus on determining relative glacier extent change. We agree that our use of 'cosmogenic nuclide dating' when referring to the bedrock core analysis is better changed to 'cosmogenic nuclide analysis'.**

Another general note is to uniformly use the term "modelling" vs "modeling". I recommend using "modelling" throughout as it appears to be more universal amongst international audiences whereas "modeling" is more generalized to the United States.

**Thank you for catching this inconsistency (due to Canada/US author team). We will change the text to 'modelling' throughout.**

Comments per line number:

31 – You reference the Little Ice Age for the first time in the body of the paper outside the abstract, but do not provide the shorthand of (LIA). Add (LIA) after you name Little Ice Age, and give the age range. (1300-1850 CE).

**Will change in text as suggested.**

36 & 39 – "Cosmogenic surface exposure dating" - consider changing to "Cosmogenic exposure analysis" because the methods applied in the manuscript to evaluate more than just the surface and dating.

**We agree with this comment when referring to the bedrock samples. While our boulder samples do have calculated exposure dates for their surfaces, the cosmogenic nuclide measurements at the bedrock sites have a more nuanced analysis.**

43 – Consider stating an approximate duration of burial required to reset or "forget" previous exposure (Hippe, 2017, Graham et al, 2019). This assumption that C-14 is completely reset during LGM burial may not be valid in some samples.

**We will include a statement on both the burial duration and subglacial erosion rate during the Last Glacial Maximum needed to fully reset the 14C inventory in the bedrock. As mentioned in Hippe (2017), 10 ka of burial with 2.4 m of subglacial erosion would reset the 14C inventory in the bedrock. We will expand in text on the evidence of other studies that give us confidence that the bedrock on Vintage Peak did experience sufficient burial and erosion during the LGM to reset the 14C inventory.**

45 – Consider referencing previous field based applications to model burial (Goehring et al, 2011, 2013; Schweinsberg et al, 2018; Pendleton et al, 2019; Graham et al, 2019; Sondergard et al, 2020; Sbarra et al, 2022; etc.)

**Thank you for these suggested studies, some of which are applied in different contexts than our study, but still have merit for consideration and discussion in our paper.**

50-52 – Include references to those studies

**We agree that we should specifically reference those studies and will include in the revised manuscript.**

Something not clearly addressed is the potential value-added information from collecting bedrock cores or subsurface samples for depth profile analysis. Why should you go to the trouble of bringing 30 kgs of equipment up the side of a mountain, scrounge for cooling water, and spend all day manually drilling to collect the core? What does the core give you that a bedrock surface sample doesn't? Please explain. Consider referencing Schaefer et al, 2016 in addition to Goehring et al 2013, Balter-Kennedy et al, 2021 and Graham et al, 2023

**This is a fair comment. We will provide additional detail and background on the potential benefit of including bedrock cores over taking just surficial samples. Namely, cores allow possible exposure and burial histories to be constrained more than surface samples alone.**

110 – Is the lidar data shown in the manuscript? Please include the figure number.

**We will include reference to Figure 5 in the manuscript text, which leverages the LiDAR data.**

116 – What type of drill was used? A portable electric hand drill or a gasoline powered system?

**We used a Shaw Tools Portable Backpack drill, which is a gasoline-powered drill. We will include this detail in the main text.**

120 – Consider changing "dating" to "measurements". The samples from the core are not being measured for ages, but concentrations that are correlated and represent the same exposure history with different shielding parameters. The concentrations from the core are what is used in the model, not the age. In general, expand on the utility of cores.

**Will change in the manuscript, as suggested, and will expand on the benefit of measuring nuclide concentrations in cores rather than surface samples alone.**

184 – Identify in the text which erratic boulder was used.

**We will specify the boulder, 17-VP-07.**

187 – State the Figure number for the two isotope plot.

**We will directly reference this plot in text.**

Additional details about the Monte Carlo simulation would be helpful.

**We agree our detail on our Monte Carlo approach is lacking and will provide additional detail on our methodology.**

194 – 50 meter ice thickness is considered sufficiently thick to cease c-14 production from muons. The authors state below (Line 228) that ice was likely 35-40 m thick. Production likely did not cease, but was probably not significant (beyond the uncertainty of the measurements) at the maximum extent. As ice thinned, and if it maintained thickness below 10 meters, it can become significant (Pendleton et al, 2019).

**While glacier ice may have been thicker than 35-40 m at Lockie's Glacier, we provide a likely minimum glacier thickness at the glacier's maximum Holocene extent. We agree that there were likely times when ice was thin enough to allow muon-produced cosmogenic isotopes, but our data are unable to quantify the duration of thin mass shielding by ice. Given the apparent exposure and burial durations at cores 2 and 3 are equivalent (within errors) we interpret the response of Lockie's Glacier to Holocene climate change to be roughly binary; the glacier was either in a advanced position close to its Holocene maximum position, or retreated to an extent similar to its modern position.**

220-221 – modelled and modeled used in the same sentence. Fix throughout.

**Will change all instances of "modeled" to "modelled" in text.**

245 Figure 4 caption – Consider removing *in situ* from the organic *in situ* 14C samples because it is more common to see *in situ* referring to *in situ* cosmogenic C-14 rather than organic. Stating organic 14C should be sufficient and likely more consistent with other literature.

**In this case, we used *in situ* to refer to the organic 14C samples, which were specifically organics that were directly killed by glacier advance and were *in situ*. While *in situ* 14C is commonly used to describe organic debris directly killed by glacier advance (e.g. Luckman, 2000), we will aim to reduce confusion and will change the text to read "... organic 14C samples from overridden wood in glacial forefields and composite moraines…"**

260 Figure 5 – This is a great figure and it is very informative that this data was made available for this study.

**Thank you!**

264 – Is there a classic canonical definition for the LIA? State clearly when the beginning is.

**While we define the LIA in the abstract (1300-1850 CE), we don't include this definition in the Introduction (Luckman, 2000). We will add this timeframe to the first sentence of the Introduction.**

266, 270, 271, 272 – The authors flip between ka (years before present) and CE (years after 0 CE). Please be consistent throughout the paper and reference events in ka or a as defined earlier in the paper. If CE is beneficial to link to other worldly events, accompany with ka and have the CE date in ().

**This is a valid critique and we will change the text to be consistent with ages as recommended.**

276 – consider replacing "some" with "approximately". Also, "returned" is very informal and colloquial, as a boulder did not deliver the age. Consider replacing with "has a calculated age of… for 10Be and …

**Agreed, will change in text.**

285 – This is a great analysis and it supports the snow data over longer time periods.

**Thank you**

Table 1 – It appears that a second row in the headers is missing.

**We apologize for this formatting error (conversion of the Google Document into a PDF causes several lines to be dropped from the table).**

If there isn't a second row missing, what is the difference between Exposure and Exposure Age?

**This is also a formatting error as mentioned above, which cut off additional text in the column header. This will be resolved in the revised manuscript.**

For the 10Be/9Be ratio, 1 sigma, Blank-corrected, Blank-corrected (assuming 1 sigma), consider using the same 10^x for each column. Cosmo tables should be formatted to easily copy and paste from the manuscript into a spreadsheet for uniform analysis. It also maintains significant figures between samples and allows the reader to quickly see which samples are orders of magnitude different. I recommend: 10^-14, 10^-15, 10^04, 10^02, respectively.

**Same as above.**

Figure 8 – Consider showing the measurement points as a box with depth to represent the integrated measurement thickness of each sample. Refer to Schaefer et al, 2016 as an example.

**We agree with this point and will modify Figure 8 as suggested.**

384 – Consider simplifying to insolation, as "solar insolation" is redundant as insolation refers to the input from the Sun.

**Agreed, will change to 'insolation'**

401 – Consider replacing "outside" with "outboard"

**Will change to "outboard".**

An assumption made throughout the paper is minimal sub-glacial erosion, and no inheritance. If there is inheritance of Be-10 in bedrock samples, is the duration of burial during the LGM and depth of erosion during the LGM sufficient enough to reset the C-14 system to background levels? If it is not, can you model to account for that?

**Our model does iterate through a range of subglacial erosion scenarios, and our data from Cores 2 and 3 are best explained with a small (but consistent in the literature) subglacial erosion rate. We will include a section in our Background that describes the time to reset any inherited 10Be or 14C under various erosion rates. As discussed in Hippe (2017), 10 ka of burial accompanied by 2.4 m of subglacial erosion removes 95% of the 14C inventory. We do think it is highly likely that Core 4 experienced ~10 ka of burial and over 2.4 m of erosion during the LGM, thereby resetting the 14C inventory. We don't have the data to evaluate the magnitude of possible 14C inheritance; however, we can modify our model to evaluate the impact of 14C inheritance on possible exposure/burial histories. We will test this effect and will include our results in either the main text or in the supplement.**

There are other assumptions made that may not be entirely valid, however, if those assumptions are not fully valid, the effect of the assumptions likely does not significantly impact the results presented. Consider reviewing all assumptions and state the effect if those assumptions are not valid.

**We aim in this paper to be explicit about each of our assumptions made and their potential impact on our conclusions. Our revised manuscript will better address how variables such as inheritance and subglacial erosion impact our results.**

**Reviewer 3 - Allie Balter-Kennedy**

This paper adds to a growing literature using cosmogenic-nuclide measurements with depth bedrock in glaciated or previously glaciated settings and among the first to do so with C-14. The manuscript not only demonstrates an approach for interpreting C-14 depth profiles, but also adds to the understanding of Cordilleran Ice Sheet (CIS) deglaciation in British Columbia and readvance of local glaciers during the Holocene. Overall, the paper is nicely presented and the scientific approach is sound. Below, I include a general comment about expanding the description and discussion of the depth profile modeling, as well as several specific comments and line edits.

**General comment:**

Because of the paper's focus on using cosmogenic-nuclide depth profiles, it would be great to see more detail included about (i) the forward model set up, (ii) the information the authors expect to gain using depth profiles, rather than just surface samples, even if the results are somewhat ambiguous, and (iii) the results of the Monte Carlo simulation.

*Information gained using depth profiles:* I really appreciate that the authors are forthcoming about the difficulties in untangling the complex exposure histories from the nuclide measurements, and therefore are conservative in their interpretation. Despite these complexities, or perhaps because of them, it could be helpful if the authors could provide a bit more information about what they expect could be gained, or what they did gain, from making C-14 measurements with depth, rather than just surface samples (i.e., why did the authors go through the trouble of collecting the cores in the first place). Similarly, the tradeoffs between exposure, burial, erosion, and other confounding factors are not necessarily straightforward to visualize, so some of my specific comments below suggest more explicit explanations of these tradeoffs to prime the reader for interpreting the MC results. Those comments are slightly more stylistic, but could make the paper accessible to a broader audience if the authors choose to implement them.

**Thank you for this comment. We agree that further background is needed to prepare the reader to understand how various exposure/burial histories, mass shielding, inheritance,**

and erosion impact the total nuclide concentrations and relative concentrations between 14C and 10Be. We sought to use depth profiles to provide additional constraints on possible exposure histories, as fewer combinations of modeled exposure, burial, and subglacial erosion produce depth profiles within measurement uncertainty along the length of our bedrock cores. We will provide additional background on how each different variables impact the net nuclide concentration in bedrock and will highlight the constraining effect of our depth profile data compared to paired surface samples alone. If there were no financial limitations, paired 14C/10Be along the entire length of the core could provide a much narrower window of acceptable exposure histories, as the measurement error of 10Be was much less than 14C.

*Forward model set up:* The reader is mostly referred to Jones et al. (2023) for details about the model set up. It would be great to see some of those details here, especially so that the reader can easily see how nuclide production rates were calculated and employed. For example, the ratio at which C-14 and Be-10 are produced increases relatively rapidly in the subsurface, yet it is unclear if muon production is included and whether the inclusion of muon-produced C-14 is important over the depths considered here (total mass shielding depth + core mass depth). If muon production is included, Equation 1 might need some updating, as right now it seems to be written only for spallation production. If it is not included, it would be nice to have that stated along with a brief explanation of why it can be omitted even in the scenarios with the higher erosion rates.

**We will include a figure, either in main text or the supplement, that shows the spallation, muon, and total nuclide production with depth for 14C and 10Be at our study site. We will also edit Equation 1, which was simplified to just show the spallation component, to show our inclusion of the muogenic component to the nuclide inventory. We agree that muon production is important in this context given an expected history of mass shielding and ambiguous erosion rates.**

*Monte Carlo results:* I found myself really wanting to see more of the MC results to evaluate the tradeoffs between exposure, burial, shielding (snow or rock), and inheritance – the authors performed 40,000 simulations, but we only see the results from four scenarios shown in the figures! The things I was most curious to see were: (i) the modeled nuclide concentrations at each sample depth compared to the measured for the best-fitting scenarios for each of the erosion rates tested, including the simulation with subaerial erosion; (ii) the range of burial and exposure durations that fit the measurements ("saved" simulations), which could be shown relatively easily by expanding Figure 9 into one panel for each core and including a contour map of X2 values for the favored erosion scenario, as well as an additional point added at the best-fitting durations for the other tested erosion rates; and (iii) the results of the simulation in which Be-10 inheritance was included, as this to me seemed like the most likely explanation for the apparent burial in Core 4. I'm sure some of this could go in the supplement, but overall I think Figures 8 and 9 in the main text could include a lot more information about the MC results.

**We agree with your suggestions for edits and additions to Figures 8 and 9, with the inclusion of the modeling results. We will edit Figure 9 to include information on the range of modeled exposure histories that fit within our measurement errors.**

**Specific comments:**

**Line 117:** "portable drill" – give more info about the drill here so others could find the same one? I assume this is a Shaw branded backpack drill? As a side note, we also struggled with the bits originally provided with that drill, so I empathize with the difficulty in retrieving longer cores!

**We will specify "Shaw Tools Portable Backpack Drill" in text.**

**Line 141:** Were the cores divided into five 5-cm-long segments, as stated, or five 5-cm-long subsamples evenly spaced downcore?

**Cores were divided into 5-cm-long segments. Since each core varied in length, some 5-cm-long sections were separated by 5 cm of unsampled core, while shorter cores had fewer unsampled sections. This difference will be more obvious in the modified Figure 8, which will show separate error boxes for each measurement, rather than interpolated errors between each measurement depth.**

**Line 131:** "presumed to have been covered by minimally erosive ice…" what is the evidence for ice being minimally erosive at that site?

**At this site, there is a lack of striations on bedrock without stepped terrain suggesting plucking. There is also no headwall at this point of the ridge; ice would have accumulated and if thick enough to flow, draped over the ridgeline, forming a (minimally erosive) flow divide at or near the site of Core 1. While we mention some of these points in the manuscript, we will include all of these points into the revised paper.**

**Line 150:** What production rate dataset and scaling methods were used to calculate the apparent exposure ages? I assume the default production rate calibration datasets, but was especially curious about scaling for both the apparent exposure ages and the forward model.

**We agree this needs to be included in the text. We used Lifton-Sato-Dunai scaling and the production rate calibration dataset based on the measurement of saturated CRONUS_A data presented in Goehring et al. (2019).**

**Line 188:** "apparent burial history" – might also be worth stating that this is the minimum/simplest/zero erosion scenario, as including erosion would yield longer total exposure/burial histories.

**We agree that apparent burial history would be appropriate here and we could/should include in text a reminder that the apparent burial history is dependent on the model parameters used.**

**Line 192:** I'm sorry if I missed it, but I was a little confused about whether the snow monitoring surveys were performed for this particular study or come from another citable study/dataset? If part of this study, could the methods for the LiDAR surveys and manual snow depth/density measurements be described in the Methods section?

**Snow depth measurements are separate and are completed by the British Columbia Government. The aerial LiDAR data was collected as part of this study by the University of Northern British Columbia/Hakai Airborne Coastal Observatory. Information on the manual snow survey methods are on the BC Government website. Background on the LiDAR acquisition and data processing are discussed in Donahue et al. (2023), published in** *Remote Sensing of the Environment*.

**Line 210-212:** I liked the consideration of inheritance as it seems a plausible explanation for the discrepancy in 10Be and 14C ages on samples outside of the LIA moraines. However, I was confused by the use of "relative 14C/10Be concentrations at Core 4". Does this just mean that 10Be concentrations in excess of the 14C age were considered inherited?

**Yes, we will clarify this statement, as we meant to say that the excess 10Be at Core 4 can be explained by ~3 kyr of apparent inheritance.**

**Line 216:** I was a little surprised that a scenario that included subaerial erosion couldn't produce exposure/burial histories that matched observations – is this just because only one subaerial erosion rate was tested? Could this exercise be described in a bit more detail and could the results be shown in the supplement?

**We tested the subaerial erosion rate of 0.001 cm/yr after a literature review that indicated that rate was reasonable in a granitic alpine setting as found at Vintage Peak** (Elkadi et al. 2022; Lehmann et al. 2020)**. However, we agree our discussion of this is brief in our initial submission. We will revise our manuscript to better show the effect of a wider range of subaerial erosion rates on our results.**

**Figure 3:** I found it a bit difficult to discern between the different line styles/colors. Is it possible for them to be slightly thicker? Also, I may have missed it in the text, but, from the caption, how is it known that Core 3 must have been ice-covered when Core 1 became exposed?

**We will re-produce the figure with thicker lines separating the various glacier extents. We interpret that Core 3 was likely ice covered with Core 1 became exposed largely from the cosmogenic data, which shows much more exposure at Core 1 than Core 3, and equivalent exposure histories between Cores 2 and 3. Based on geomorphology in the**

**cirque, we also expect that ice formed at Core 1 would have been more peripheral to the main flowline of the glacier.**

**Line 292:** If there is ~2 ka inheritance in the 10Be age, shouldn't the age of deglaciation be constrained to 14.5–9.7 ka (the C-14 age), rather than 11.6 ka (the Be-10 age)? If so, this may need to be updated elsewhere, too.

**This is a fair point and we agree that the C-14 age is the most conservative exposure age for Core 4. We will present this more comprehensive range of possible deglaciation ages at Core 4 here and at other locations in text (i.e. Abstract, Conclusion).**

**Figure 7:** I didn't quite understand the normalization to the Core 4 concentrations, rather than the local production rate at each site, since normalizing to the site-specific production rate accounts for differences in concentration due to production rate differences, allowing concentrations to be compared. I think a clearer explanation of why normalization to the Core 4 concentrations was chosen either here or in the text would be helpful!

**The concentrations presented in Figure 7 were scaled to the site-specific production rate relative to Core 4, in order to preserve the magnitude of nuclide concentrations in each core. The equations used for this normalization was:** $Norm\ Conc\ (N)\ =\ N_x / (1 - \frac{(P_x - P_{Core4})}{P_{Core4}})$ **, where $N_x$ is the measured nuclide concentration ($^{10}$Be or $^{14}$C), $P_x$ is the site-specific production rate, and $P_{Core4}$ is the production rate at Core 4.**

**Table 1**: Could this include the 14C surface concentrations and apparent exposure ages as well? If not, could those be included somewhere in the main text? Also, I believe this will be fixed during typesetting, but the headers were cut off in some columns.

**Yes, there were some formatting issues with our initial submission, headers will be included in typesetting. We agree having the 14C ages in the main text would be helpful. We will modify Table 1 to include the 14C apparent exposure ages for the bedrock surface samples and 17-VP-07 erratic.**

**Figure 8:** It's nice to see the variation of 14C concentrations with depth in the cores next to each other, but I do think this figure could be improved. First, because a uniform density is assumed throughout the core, could you add a second y-axis that as linear core depth for ease of interpretation? Second, the use of a line plot here makes it seem as if continuous measurements are made, which is misleading. Could you instead use points with error bars at the actual sample depths to represent measured concentrations and their uncertainties? 14C could also be added to the x-axis label.

**As addressed above, we agree with these suggestions and will edit Figure 8 ss recommended.**

**Section 4.3:** This relates to my general comment, but when describing the change in nuclide concentration with core depth, it might be helpful somewhere to state or show what the variation of nuclide production rate with depth looks like (i.e., do these look as if they are affected by erosion?)

**As mentioned above, we will provide additional background (which will likely include a new figure) that shows the expected nuclide concentration with depth (both spallation and muon-produced), including the influence of mass shielding, erosion, and inheritance on the expected nuclide profile.**

**Section 4.4:** It would be helpful either here or somewhere above to prime the reader with an understanding of how shielding contributes to changes in the modeled exposure/burial histories. For example, the fact that "increasing the subglacial erosion rate notably increased the modelled exposure duration and decreased the burial duration" (Lines 350-351) is an expected outcome that could be described to help guide the reader through the results. Somewhere could be worth saying that if there is shielding (rock, snow, till), the measured nuclides were produced deeper in the rock column where production is lower, so more exposure is needed to reach the measured concentrations. Because of the relatively short 14C half-life, however, longer exposure also drives a lower 14C/10Be ratio, so at least a portion of the lower measured ratio can be explained by longer exposure, requiring less burial. This is similarly why apparent burial can also be interpreted as additional shielding.

**We agree with this point and the need for providing additional background on how the measured $^{10}$Be and $^{14}$C concentrations, and their relative ratios, are influenced by mass shielding, erosion, and inheritance.**

**Lines 372-373 and 379:** How much till cover is needed to account for the 3 ka burial for Core 4? I agree with the authors that till cover is a less likely explanation than inheritance, but it would be helpful to see the 14C apparent age and the results of the inheritance exercise to evaluate that more fully.

**Here, we assume that there was sufficient till cover to effectively shut off nuclide production at Core 4. Assuming till has a density of 2.4 gcm$^{-3}$, it would take ~1.9 m of till over Core 4 to have the same mass depth as 50 m of 0.9 gcm$^{-3}$ glacier ice (more than enough depth to reduce nuclide production levels below detectable limits). Testing the effect of thin till or ice cover at Core 4 is beyond the scope of this modelling study. We will include this information in the revised version of the manuscript.**

**Line edits:**

**Line 17:** "shallow bedrock cores" – add (<0.6 m depth) for context?

**Agreed, will add to text.**

**Line 19:** Sentence starting "Vintage Peak became uncovered" – add something to indicate these are results from this study?

**Will change the sentence to start with "We found that Vintage Peak became…"**

**Line 59** "even cirque glaciers … experienced cover by readvance" – change to "glaciers readvanced" or "bedrock experienced cover as glaciers readvanced", as it's the bedrock not the glaciers that are experiencing cover.

**We agree with this change and will edit accordingly.**

**Figure 1:** This is a really nice figure overall, but I was struggling to see some of the text and symbols. Could those be slightly larger, including the start in the inset?

**We will increase the text size to improve readability.**

**Line 100:** "six boulders represent samples from Lockie moraine" – I wasn't quite sure what "represent" means here – maybe just, we sampled six boulders from lockie moraine …"?

**We agree with the suggested clarification.**

**Line 149:** I believe the paper cited here as Balco et al. (2022) was actually published in 2023 (although the preprint on EGU sphere was posted in 2022!)

**Good catch! We will update the citation with the 2023 peer-reviewed article.**

**Line 266**: maxima, not maximum

**Will change to 'maxima'**

**Line 239:** "10Be date to…" felt slightly awkward/colloquial to me, could this be changed to "have 10Be ages of…"

**Agreed, will change as suggested.**